# Reflections about Blockchain in Health Data Sharing: Navigating a Disruptive Technology

**DOI:** 10.3390/ijerph21020230

**Published:** 2024-02-16

**Authors:** Ana Corte-Real, Tiago Nunes, Paulo Rupino da Cunha

**Affiliations:** 1Clinical and Academic Centre of Coimbra, 3004-531 Coimbra, Portugal; tiago.nunes@fmed.uc.pt; 2Faculty of Medicine, University of Coimbra, 3000-370 Coimbra, Portugal; 3Centre for Informatics and Systems of the University of Coimbra, Department of Informatics Engineering, University of Coimbra, 3030-790 Coimbra, Portugal; rupino@dei.uc.pt

**Keywords:** blockchain, health, legal medicine, migrants

## Abstract

A comprehensive analysis was performed, considering blockchain technology (BT) properties in digital health, addressing medicolegal, privacy, and regulatory considerations. Adherence to personal data protection and healthcare regulatory guidelines were analyzed and compared for GDPR (Europe), HIPAA (United States), CCPA (California), PIPEDA (Canada), the Privacy Act of 1988 (Australia), APPI (Japan), and LGPD (Brazil). Issues such as health systems, strengthening and aligning policy orientations and initiatives, and emphasizing the role of data analysis in shaping health policies were explored. The study addressed conflicts between the legal frameworks and blockchain, comparing and suggesting solutions like the revision of laws and the integration of compliance mechanisms. Additionally, it sought to enhance IT-health literacy by integrating the healthcare and legal domains. Ongoing collaboration between legal, health, and IT experts is essential for designing systems that effectively balance privacy rights and data protection while maximizing the benefits of disruptive technologies like blockchain.

## 1. Introduction

In 2015, the United Nations Members identified 17 sustainable development goals for 2030 [1]. To make the 2030 Agenda a reality, the European Commission (EU), in May 2023, reorganized and interconnected specific goals and their deliverables into strategy lines for universal health-related targets [2]. External action, including addressing universal health coverage and addressing root causes of ill-health like poverty and social inequalities (known as “health-in-all-policies”), was emphasized [1,2]. Consequently, communication skills were targeted to facilitate global action with international standards [3]. Simultaneously, the World Health Organization (WHO) has been drawing attention to the growing trend of forcibly displaced people worldwide caused by globalization, low transportation costs, economic pressures, demographic trends, environmental degradation, violence, armed conflicts, and human rights abuse [3]. These migration flows present a significant challenge to human rights and health systems in providing equitable access to healthcare and managing health data [4].

The Global Consultation on Migrant Health emphasizes the need to strengthen health information systems to collect and disseminate migrant health data [3,4]. According to the WHO, inclusive health systems for migrants improve public and global health outcomes for all [3]. Nevertheless, the WHO points out barriers to implementing universal health policies, including national policies for migrants and social and cultural values [3]. The recent COVID-19 pandemic has underscored the importance of creating a global space of solidarity and justice that aligns with Sustainable Development Goal 17 (SDG 17). It tested the resilience of health systems in delivering essential services to safeguard people’s lives [1,2,5]. Efficiency and transparency in health surveillance can benefit significantly from consolidating world population data. However, since confidential information is at stake, it is essential to consider international guidelines and regulations for health databases.

In a global context, personal rights must be considered [5] and recognized under the legitimate framework of each country. This entails upholding the principles of autonomy and self-determination while emphasizing justice and non-discrimination practices [6,7,8,9,10,11,12,13,14,15]. In the context of sensitive medical data, individual rights are recognized by healthcare standards. EU Member States play a pivotal role in global health policy analysis. In Europe, the General Data Protection Regulation (GDPR) establishes a unified framework for EU citizens [6]. It emphasizes specified and explicit purposes, aligning objectives, and empowering individuals to make informed decisions through personal consent. Key considerations generally include (i) obtaining consent for specific data processing purposes; (ii) addressing contractual necessities, including data subject requests; (iii) fulfilling legal obligations imposed on data controllers; (iv) safeguarding vital interests; (v) fulfilling public tasks or exercising official authority; and (vi) pursuing legitimate interests. The analysis and management of sensitive data should align with principles of individual privacy, transparency, and accountability. This involves granting data access, providing information about procedures, and conducting impact assessments.

The global context just described makes it essential to reflect on health data issues concerning personal rights regulation and laws. Furthermore, it is crucial to understand how promising technologies like blockchain can be aligned with legal requirements.

In the remainder of this paper, Section 2 compares data major data protection regulations as they apply to health data; Section 3 discusses policy orientations and initiatives for strengthening health systems; Section 4 introduces blockchain technology in the discussion, to assess whether it fulfills the requirements for Health Systems strengthening; Section 5 furthers the discussion, with an analysis of the impact of blockchain technology on law and human rights; Section 6 identifies potential conflicts between blockchain and GDPR and offers possible solutions; and Section 7 discusses future research directions, before we conclude in Section 8.

## 2. Comparison of Data Protection Regulations

The European Union’s General Data Protection Regulation (GDPR) is considered a landmark in this kind of legislation that has influenced similar initiatives around the world. This section compares it with legal frameworks from other significant jurisdictions.

The United States adheres to the Health Insurance Portability and Accountability Act (HIPAA), a healthcare standard that outlines various legitimate grounds for processing protected health information, including treatment, payment, healthcare operations, and public health activities [11]. Additionally, consent requirements can differ depending on state laws, with some states, such as California, imposing additional regulations on medical data privacy and consent, exemplified by the California Consumer Privacy Act (CCPA) [7]. CCPA is the most similar data protection framework to GDPR, granting individuals the right to access their data, request its deletion, and opt out of data processing. It strongly emphasizes transparency, requiring organizations to provide clear and easily accessible privacy notices that detail data collection, usage, and sharing practices. CCPA encourages the principle of data minimization, meaning that organizations should only collect and retain the necessary personal data for specific purposes. Furthermore, it highlights the importance of organizational accountability in handling personal data and implementing appropriate security measures to protect it from unauthorized access or breaches. Both GDPR and CCPA possess extraterritorial reach, extending their jurisdiction to organizations located anywhere in the world as long as they process the personal data of individuals from Europe or California, respectively.

In Canada, medical data processing is governed by the Personal Information Protection and Electronic Documents Act (PIPEDA) at the federal level, complemented by various provincial health privacy laws [12]. Australia’s Privacy Act of 1988 regulates medical data processing. These regulations encompass obtaining consent, fulfilling contractual obligations, protecting vital interests, performing public functions, and pursuing legitimate interests [14]. Japan protects medical data and individual rights through the Act on the Protection of Personal Information (APPI) [16] and the Act on Assurance of Medical Care for Elderly People [17]. In Brazil, the Lei Geral de Proteção de Dados (LGPD) governs the processing of personal data, including medical data [18].

The use of digital technologies was notably accelerated globally during the COVID-19 pandemic with the vaccination certificate and telehealth [15]. The EU is at the forefront of establishing a global platform to support medical action during future global health emergencies [2]. It aims to improve communication, information access, and personal identification within public and national health services and support migration flows and networks [17,18,19]. In addition, managing medical records in a digital format, such as electronic health records (EHR), will be approached in an inter-organizational manner, assessed at a global level, rather than being confined to internal processes of health institutions.

Medical institutions have increasingly turned to websites and mobile applications to access and manage their own and patient-generated health data [20]. Advances in information systems have led to the reconfiguration of technological procedures. More recently, blockchain, a type of distributed ledger technology (DLT), holds promise in securely recording and sharing health data across a decentralized network of peers, offering real-time worldwide access to users both within and outside national health systems. Furthermore, blockchain utilizes cryptographic mechanisms to ensure historical data immutability and integrity [19,21,22]. Considered use of blockchain and ancillary technologies can lead to private, secure, and resilient health information systems compliant with relevant laws and regulations.

This study aims to comprehensively analyze blockchain technology (BT) within the context of digital health, addressing medicolegal, privacy, and regulatory considerations [23,24,25,26]. The primary focus is on ensuring personal data protection and adhering to healthcare regulatory guidelines while promoting transparency and a coordinated response to emerging trends. Additionally, the study seeks to enhance the knowledge and literacy of stakeholders in both the healthcare and legal domains.

## 3. Health Systems Strengthening—Policy Orientations and Initiatives

For both public and private healthcare institutions, prioritizing innovation is essential for enhancing their standing in the healthcare market and ensuring long-term sustainability. Pursuing an improved focus on health cannot be an individual effort; it must be developed globally, in line with the universal right to healthcare.

The European Union (EU) contributes to advancing public health policies by engaging with stakeholders [2]. Initiatives such as the EU Health Policy Platform and the Commission’s expert group on public health facilitate discussions on public health issues, knowledge sharing, and the dissemination of best practices [3].

The EU promotes health systems strengthening (HSS) as a coordinated objective among its member states. As defined by the World Health Organization (WHO), HSS is the process of identifying and implementing policy and practice changes within a country’s healthcare system to better address its health-related challenges. It encompasses various initiatives and strategies that enhance the functions of the healthcare system, resulting in improved health outcomes through better access, coverage, quality, or efficiency [3].

The WHO outlines six key building blocks for HSS: health service delivery, health workforce, health information systems, access to essential medicines, health systems financing, and leadership and governance. Information systems (IS) play a pivotal role in the journey towards HHS; they facilitate the implementation of health regulations, such as the International Health Regulations (IHR), across inter-sectorial fields, spanning from healthcare to research institutions and emerging technologies [2].

The significance of humanitarian health assistance became increasingly apparent during the pandemic, emphasizing the necessity for a comprehensive response. In pursuit of improved global health outcomes, the EU has identified three critical factors: digitalization, research, and security [22]. These enablers are geared toward ensuring more equitable access to healthcare, strengthening disease surveillance and detection, and adopting a comprehensive “One Health” approach, which integrates environmental, animal/plant, and human health concerns [4]. These elements are indispensable for effective external policies in the complex geopolitical landscape, relying on international partnerships founded on co-ownership and co-responsibility, a unified and influential voice, and innovative financing mechanisms.

Both the WHO and EU initiatives are committed to a holistic perspective in legitimizing secure health data sharing. Health data analysis provides a framework for shaping health policies. The legal system plays a pivotal role in safeguarding fundamental and patient rights while establishing the statutory regulations needed to implement these principles and rights. By examining the interplay between the General Data Protection Regulation (GDPR), the Human Rights Charter, and technology implementation, we can thoroughly assess the advantages of technology within an adapted regulatory framework.

## 4. Can Blockchain Technology Fulfill the Needs of Health Systems Strengthening?

In the field of medical data, blockchain-based information systems play a crucial role in breaking free from the constraints of a single healthcare provider, making data accessible on a global scale to various health stakeholders. They address the limitations inherent in traditional healthcare systems, where patient data are usually centralized. A distributed system must, however, ensure that digital data remain robust, resilient, and protected against issues during transmission or unauthorized access. A top priority for such a system is security, trustworthiness, and safeguarding data integrity. Additionally, it should empower individuals in the management of their data.

Blockchain technology (BT) meets these requirements. BT is a type of distributed ledger technology (DLT) operating in a peer-to-peer network of nodes. Each node in the network maintains a real-time copy of the complete data, which is securely stored in blocks cryptographically linked together to form a chain. Encryption and authentication mechanisms can also be used to guard against unauthorized access, fraud, and tampering. Consensus algorithms ensure the legitimacy of data added to the ledger.

In the healthcare context, healthcare providers can access and contribute to the blockchain securely by utilizing cryptographic keys. This facilitates the establishment of a transparent, auditable, reliable, and tamper-resistant medical record history and the possibility to grant or revoke selective access to health information. BT can also be used to assign secure and portable digital identities, namely, to individuals lacking official documents.

## 5. Blockchain Technology’s Reflections in Law and Human Rights

Blockchain technology represents a paradigm shift in the secure management, storage, and sharing of data. However, it raises critical concerns about compliance with the law and human rights as they are currently applied to issues related to health data. The fundamental concept of blockchain technology will be discussed—decentralized data registries that improve sharing and interoperability in accessing information, promoting better communication between stakeholders and reducing information disparities in line with immutability and data integrity [26].

The essential premise of medical data collection, identified as the legal basis for processing personal data, is expressed as the rights of the individual and involves obtaining consent and ensuring legal compliance from all relevant parties. Free and informed consent is obtained by the professional working in the healthcare institution. This process has been debated and established, with specific and strict requirements, in GDPR (Europe), CCPA (USA), APPI (Japan), and LGDP (Brazil) and as a recommendation in HIPAA. However, this process may have more flexibility under PIPEDA and the Privacy Act 1988. GDPR grants individuals certain rights over their personal data, including the right to access, rectify, erase, and restrict data processing. However, the immutability of the blockchain, which ensures that once data are recorded, they cannot be altered, poses a challenge in fulfilling the right to erasure (also known as the “right to be forgotten”), which is not necessarily considered in the regulations of all countries (Table 1). Blockchain’s immutability may conflict with the GDPR’s, PIPEDA’s, and LGPD’s requirement to delete personal data on request.

Protecting and regulating medical data involves three conditions that are mandatory or recommended in international regulations: (1) the legal basis for data processing and governance; (2) data protection by design; and (3) data minimization and purpose limitation.

The various legislations generally share the requirement that organizations have a legal basis for processing personal data when obtaining consent, fulfilling a contractual obligation, or complying with legal obligations. In the context of blockchain, where multiple participants contribute and validate data, determining the legal basis for processing and obtaining appropriate consent from all parties involved can be a challenge, which is reflected in data governance [27].

Internal and external auditability should be highlighted, ensuring transparency and control of transactions or accountability. Regular compliance with quality standards is subject to audits to identify vulnerabilities and ensure adequate quality measures. Smart contracts, a unique feature of blockchain technology, and non-fungible tokens (NFTs) offer advanced capabilities for establishing compliance mechanisms and auditing processes. Smart contracts, which are self-executing and tamper-proof, can autonomously record and enforce access permissions, data modifications, and transaction history on the blockchain network. Non-fungible tokens, by representing unique digital assets, provide a means of verifying and authenticating users’ identities and their corresponding actions in the system. Together, these tools provide a comprehensive framework for transparently tracking and verifying data access, modifications, and transactions, thereby reinforcing accountability and ensuring strict compliance with legal and ethical standards, including those related to human rights.

The responsibilities and monitoring of medical data align with the law, ethics, and healthcare regulation standards. Legal frameworks and privacy regulations should be considered when designing and implementing blockchain systems. The control of personal data and their confidentiality follows governance and consent management strategies. An effective governance framework includes establishing policies, guidelines, and protocols for data access, storage, and sharing, as well as mechanisms for resolving disputes, enforcing compliance, and managing system upgrades and enhancements. This network monitoring process can implement access controls and permission mechanisms, following different levels of access by nodes based on their roles and responsibilities. This ensures that authorized entities with the necessary permissions can access personal data. Protection or regulatory standards must be guaranteed. Technology implementation must comply with existing regulations, such as data protection laws [6,7,8,9,11,12,14,16,17,18] and healthcare standards [10,13,15].

Concerning data protection by design, the GDPR and LGPD promote privacy by a supervisory authority, or a local one, for PIPEDA and AAMCEP, which means that these considerations must be integrated into the configurations of systems that process personal data. However, blockchain systems operate with transparent and immutable data, which can conflict with the concept of privacy by default since personal data are made accessible to the consortium by design.

When it comes to data minimization and purpose limitation, these principles are generally emphasized, requiring organizations to record only the data necessary for their objectives. Some blockchain-based systems store only hashes of the data in the ledger to ensure its integrity, while the data are stored off-chain.

## 6. Addressing GDPR and Blockchain Conflicts: Potential Solutions

Resolving the conflicts between the GDPR and the immutability of blockchain technology requires careful consideration and potential adaptations. Four strategies can be explored: revision of laws and regulations, compliance mechanisms, use of hybrid network architectures, and governance frameworks for privacy assessments.

First, consider revising laws and regulations to avoid the compulsory deletion of medical data. Regulations in the United States (HIPAA), California (CCPA), Australia (Privacy Act 1988), and Japan (APPI) already do not necessarily contemplate the deletion of medical data. Adequate data management can balance the rights of the individual and the impact on the community. On the one hand, preserving medical data promotes health as an individual good thanks to better clinical history. On the other hand, it promotes health as a universal good, thanks to aggregate analyses that benefit the population (e.g., distribution of vaccines and allocation of health resources) and the advancement of knowledge and research (e.g., new treatments and medicines). However, preserving medical data should not endanger individuals’ privacy. One possibility is to resort to anonymization. Responsibility in data governance is an opportunity to put ethical aspects into practice, adjusted to the type of data (e.g., the issues raised by genetic data are different from imaging data) [27]. The institutions that store the data must follow ethical guidelines and standards (e.g., transparent policies for safe and responsible storage) in accordance with leges artis. When considering this issue, only theoretical concepts can be put forward, and basic moral principles can be modified to adapt to the new challenges of the digital age. As Hallammaa and Kalliokoski point out, it is impossible to conduct research without first examining the project from an ethical perspective [28,29]. In addition to its focus on the individual, ethics in health and well-being is also concerned with the impact of the technologies underlying care. As with artificial intelligence (AI), the use of blockchain should be governed by an ethical framework and processes to define strategic objectives and pursue results.

Second, consider integrating compliance mechanisms, auditing processes, and privacy-enhancing technologies in the context of blockchain use. One crucial strategy involves refraining from directly storing personally identifiable information on the blockchain and leveraging privacy-enhancing technologies, like encryption or pseudonymization, to protect personal data while upholding the integrity of the blockchain.

Third, consider using hybrid architectures to strike a balance between data integrity and blockchain’s immutability [30]. By combining blockchain with off-chain storage, the latter can be used to store or delete sensitive data, while the former can be used to store hashes of those data—“digital fingerprints” that attest to the integrity of the off-chain data.

Lastly, consider governance frameworks to ensure compliance with data protection laws and mitigate privacy risks, for example, by establishing agreements among blockchain participants to guarantee the effective handling of the data subject’s rights and requests and conducting privacy impact assessments to identify and address privacy risks associated with blockchain implementations systematically.

## 7. Future Research Directions

This paper compared data protection regulations and frameworks of six major jurisdictions as they apply to health data. However, the work can be extended to achieve an even more comprehensive global understanding by considering additional relevant players. For example, the United Kingdom (the UK GDPR), South Korea (the Personal Information Protection Act), and Singapore (the Personal Data Protection Act that regulates the collection, use, and disclosure of personal data by organizations in the Asia–Pacific region).

Additionally, the potential of blockchain technology to challenge the concept of a centralized health database can be explored in a proof-of-concept work and validated or implemented in real scenarios while complying with global data protection regulations and frameworks.

## 8. Conclusions

Blockchain technology can potentially address the challenges of traditional healthcare systems by providing decentralized global accessibility. To ensure adherence to personal data protection and healthcare regulatory guidelines, a comparison was made between GDPR (Europe), HIPAA (United States), CCPA (California), PIPEDA (Canada), Privacy Act of 1988 (Australia), APPI (Japan), and LGPD (Brazil), highlighting consent requirements. However, there are conflicts between regulatory guidelines and blockchain characteristics, underscoring the need for ongoing discussions and collaborations between legal and technical experts. Potential solutions, including regulatory revisions and governance frameworks, need exploration to address concerns around compliance with laws and human rights. Thoughtful system design is necessary to balance privacy rights, data protection requirements, and the unique characteristics of blockchain technology.

## Figures and Tables

**Table 1 ijerph-21-00230-t001:** Provides a comparison of Europe’s General Data Protection Regulation with similar initiatives in different regions such as the United States (HIPAA), California (CCPA), Canada (PIPEDA), Australia (Privacy Act 1988), Japan (APPI), and Brazil (LGPD) [6,11,12,14,16,17,18]. Legend: GDPR = General Data Protection Regulation, Europe; HIPPA = Health Insurance Portability and Accountability Act, USA; CCPA = California Consumer Privacy Act; PIPEDA = Personal Information Protection and Electronic Documents Act, Canada; APPI = Act on the Protection of Personal Information, Japan; AAMCEP = Act on Assurance of Medical Care for Elderly People, Japan; LPGPD = Lei Geral de Proteção de Dados, Brazil; x = required; - = Absent information.

		Europe	USA		Canada	Australia	Japan		Brazil
		GDPR	HIPAA	CCPA	PIPEDA	Privacy Act 1988	APPI	AAMCEP	LGPD
Public target		European Union and EU citizens	Health institutions and health insurance activities in EUA	Businesses in California and California citizens	Commercial activities in Canada	Government agencies and private-sector organizations	Institutions in Japan	Elderly residents in Japan	Institutions in Brazil
Organization sector		All	Health	Business	Private	Private and Public	All	Geriatric Institutions and stakeholders	All
Personal Consent	Requirements	Specific	Informed and recommended for sensitive data	Specific	Flexible	Specific	Flexible	Specific	Specific
		Informed and explicit for sensitive data		Informed and explicit for sensitive data	Mandatory for sensitive data	Informed and explicit for sensitive data	Informed and explicit for sensitive data	Mandatory for sensitive data	Informed and explicit for sensitive data
	Timeframe	Before collecting, using, or disclosing medical data	Not required, but recommended	Before collecting, using, or disclosing medical data		Not required, but recommended	Not required, but recommended	x	Not required, but recommended
	Access data	x	x	x	x	x	x	x	x
	Request data Correction and deletion	x	x	x	x	x	x	x	x
	Withdraw Consent	x	-	-	x	-	-	-	x
	Transparency	x	x	x	x	x	x	x	x
	Accountability	x	x	x	x	x	x	x	x
	Impact assessments	Specific risks Required	x	Specific risks Required	Specific risks Recommended	x	x	x	Specific risks Recommended
Data Breach Notification (within Specific Timeframes)		Supervisory authority			Local authority	No specific authority	No specific authority	AAMCEP’s authority	Supervisory authority
		Fines for non-compliance			Non-compliance, reputational damage, and legal impact	Fines for non-compliance			

## Data Availability

The original contributions presented in the study are included in the article, further inquiries can be directed to the corresponding authors.

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
