# Peer review of "Reflections about Blockchain in Health Data Sharing: Navigating a Disruptive Technology"

_ijerph, 2024, doi:10.3390/ijerph21020230_

Round 1

Reviewer 1 Report

Comments and Suggestions for Authors

1.       Provide authorized report for “WHO points out barriers to implementing universal health policies, including national policies for migrants and social and cultural values”

2.       So much lengthy introduction.

3.       Research gaps not provided.

4.       Authors fails to stream line the objectives of the paper.

5.       Which kind of authorized medical records in a digital format the authors are talking about? Are they available publically?

6.       Table captioning are not proper.

7.       How the authors decide to consider - revision of laws and regulations, compliance mechanisms, use of hybrid network architectures, and governance frameworks for privacy assessments for consideration and potential adaptations?  Only these are sufficient?

8.       Conclusion not matched with the work presented in the paper

9.       Future work must be highlighted

10.   Many references are not accessible.

Author Response

Dear Reviewer,

You will find a detailed description of how we addressed each comment in the document attached. As the comments and answers intersected on similar topics, I shared them with all authors. The authors did our best to address the reviewers' concerns in the allotted time frame. We have segmented each reviewer's comments into sections addressing specific issues to ensure their systematic coverage.

Kind regards,

The authors

Reviewer 2 Report

Comments and Suggestions for Authors

The authors have made good efforts to provide an analytic reflection on the emergence of blockchain technology in healthcare. Overall, with the advancement of technology and the growing need of handling vast quantity of health data, the reflective article comes as a timely contribution to the contemporary development of healthcare around the world. Insights are invaluable to the evolving landscape of healthcare, in particular in the post-pandemic era. Overall, I see good values of current work and have the following suggestions:

1.

The article type is indicated as “Communication” on the first page. To me, it comes as a perspective considering the current context of the manuscript. Would consider adopting the current work as a perspective article?

2.

On page 2-4, the analysis based on the cases in various countries are acknowledged. Would the authors consider highlighting the reason for including the selected examples. Do they share some common characteristics in the health system or cases are based on geographic diversity to provide an overview of analysis? It would be good to briefly describe the features behind for considering these cases.

3.

I suggest on p.8, at the last part of the discussion, i.e. before the “Conclusion” section, can explore more details on the future directions and recommendations to the issues observed, in particular there are some good reflections and analysis on the challenges of blockchain technology in the previous paragraphs. The added insights can be further extended to providing a framework for future research directions.

4.

The introduction section (p.1) starts well with the mentioning of UN SDGs. It would be good for indication of the interconnections of health with other components / systems in sustainable development and public well-being. would consider support the background and explanations by adding recent references. The authors may consider details from the following or other relevant discussions:

Chiu, W. K., & Fong, B. Y. F. (2023). Sustainable Development Goal 3 in Healthcare. In Environmental, Social and Governance and Sustainable Development in Healthcare (pp. 33-45). Singapore: Springer Nature Singapore.

Khan, F. A., Asif, M., Ahmad, A., Alharbi, M., & Aljuaid, H. (2020). Blockchain technology, improvement suggestions, security challenges on smart grid and its application in healthcare for sustainable development. Sustainable Cities and Society55, 102018.

Author Response

Dear Reviewer,

You will find a detailed description of how we addressed each comment in the document attached. As the comments and answers intersected on similar topics, we shared them with all authors. We did our best to address the reviewers' concerns in the allotted time frame. We have segmented each reviewer's comments into sections addressing specific issues to ensure their systematic coverage.

Kind regards, 

The authors

Reviewer 3 Report

Comments and Suggestions for Authors

The followings are my observations and suggestions while reviewing this paper.

 The paper showcases the the United Nations Members identified 17 sustainable development goals for 2030 which they presented in 2015 and the author cited that as well. But the question arise the innovative goal and objective of this journal paper. The said journal expect a original research article or a review of existing technology where scientific comparison should be presented. This article looks like a report not like a scientific journal.

Hence, this report which is already published by corresponding authority as I mention above is not suitable for a original scientific research or Review.

Comments on the Quality of English Language

The followings are my observations and suggestions while reviewing this paper.

 The paper showcases the the United Nations Members identified 17 sustainable development goals for 2030 which they presented in 2015 and the author cited that as well. But the question arise the innovative goal and objective of this journal paper. The said journal expect a original research article or a review of existing technology where scientific comparison should be presented. This article looks like a report not like a scientific journal.

Hence, this report which is already published by corresponding authority as I mention above is not suitable for a original scientific research or Review.

Author Response

(The authors gave the same response as above.)

Round 2

Reviewer 1 Report

Comments and Suggestions for Authors

Paper has been updated as per the comments

Reviewer 3 Report

Comments and Suggestions for Authors

The paper improves a lot due to review process. Most of the review comments are address very well. I have no further objection. The paper could be accepted.

Comments on the Quality of English Language

The paper improves a lot due to review process. Most of the review comments are address very well. I have no further objection. The paper could be accepted.